# EGALA: EFFICIENT GRADIENT APPROXIMATION FOR LARGE-SCALE GRAPH ADVERSARIAL ATTACK

## ABSTRACT

Graph Neural Networks (GNNs) have emerged as powerful tools for graph representation learning. However, their vulnerability to adversarial attacks underscores the importance of gaining a deeper understanding of techniques in graph adversarial attacks. Existing attack methods have demonstrated that it is possible to deteriorate the predictions of GNNs by injecting a small number of edges, but they often suffer from poor scalability due to the need of computing/storing gradients on a quadratic number of entries in the adjacency matrix. In this paper, we propose EGALA, a novel approach for conducting large-scale graph adversarial attacks. By showing the derivative of linear graph neural networks can be approximated by the inner product of two matrices, EGALA leverages efficient Approximate Nearest Neighbor Search (ANNS) techniques to identify entries with dominant gradients in sublinear time, offering superior attack capabilities, reduced memory and time consumption, and enhanced scalability. We conducted comprehensive experiments across various datasets to demonstrate the outstanding performance of our model compared with the state-of-the-art methods.

## 1 INTRODUCTION

Graphs are widely used to represent interrelations in various domains of the modern world, such as social networks, protein-protein interaction, and knowledge graphs. Extensive research has been conducted on Graph Neural Networks (GNNs) to effectively capture graph representations (Hamilton et al., 2017). While GNNs demonstrate remarkable performance in graph representation learning, they are also vulnerable to adversarial attacks. Specifically, it has been noted that perturbing the adjacency matrix by adding just a few edges can significantly alter the predictions of GNNs (Waniek et al., 2018; Dai et al., 2018; Zügner et al., 2018). This finding underscores the importance of robustness of GNNs in environments with potential malicious users, thereby necessitating the development of highly efficient attacking algorithms for red-teaming GNN models.

In this study, we consider structural attacks on GNNs, where the attacker aims to interfere the prediction of GNNs by adding or removing a few edges in the graph. Although several attacking methods have been proposed in the literature (Bojchevski & Günnemann, 2019; Zügner et al., 2020; Deng et al., 2022), most of these approaches have limited scalability due to a fundamental challenge in structural attack – when conducting graph adversarial attacks on edges, one must consider all $N^2$ entries in the adjacency matrix, where $N$ represents the number of nodes in the graph. For larger graphs, computing gradient on all $N^2$ edges become prohibitive, making them impractical for large graphs. To tackle this issue, recently Geisler et al. (2021) introduced a scalable structural attack algorithm by only considering a random subset of $\mathcal{O}(b)$ edges at each attacking step, where $\mathcal{O}(b) \ll \mathcal{O}(N^2)$. With this approximation, they are able to compute the gradient on all $\mathcal{O}(b)$ entries and select the top ones to attack at each step. The improved scalability also enables them to conduct global attacks on large graphs, showing it is possible to perturb a few edges to downgrade the overall prediction preformance of GNNs.

Despite being an efficient approximation, there is no guarantee that vulnerable edges will be located in the sampled block, potentially resulting in suboptimal outcomes for this attack. Consequencely, as the graph size grows, they need to sample larger blocks correspondingly in order to maintain the attack performance (Geisler et al., 2021). Computing exact gradients for all elements within the block imposes a significant computational burden and a high memory cost.

In response to these limitations, we propose our model EGALA, an innovative approach for efficiently conducting large-scale graph adversarial attacks. Given that the adjacency matrix is inherently discrete, our strategy for edge addition/deletion during coordinate descent doesn't necessitate precise gradient values across all $N^2$ matrix entries. Instead, we focus exclusively on identifying the $m$ entries with the most significant gradients. Building upon this concept, we approximate the gradient of loss across the entire adjacency matrix and convert it to the inner product of two $N$-by$-d$ matrices, where $d$ represents the latent feature dimension. With this approximation, the top $m$ entries can be selected by efficient Approximate Nearest Neighbor Search (ANNS) algorithms, which runs in *sublinear time* and does not require full scan through all $N^2$ entries (Johnson et al., 2019; Malkov & Yashunin, 2018). We then implement a clustering-based ANNS algorithm on these two $N$-by-$d$ matrices to efficiently find top elements at each iteration for conducting our attack.

To rigorously assess the effectiveness of our proposed model EGALA, we conducted comprehensive experiments across various datasets, comparing with the state-of-the-art PRBCD and GRBCD models proposed in Geisler et al. (2021). The results demonstrate that EGALA exhibits significantly higher efficiency compared to previous methods, while also achieving a slightly enhanced attacking performance.

Our contributions can be succinctly summarized as follows:

• We propose a novel approach EGALA for conducting graph adversarial attacks through efficient gradient approximation. The key innovation of EGALA lies in its successful reconfiguration of the computation of loss gradients across the entire adjacency matrix as the inner product of two $N$-by-$d$ matrices. This enables the identification of entries with the most significant gradients in sublinear time through ANNS and makes the attack scalable to large graphs.

• Our model consistently outperforms baseline models across vairous datasets, showcasing its effectiveness in conducting graph adversarial attacks. Its consistently outstanding performance across diverse GNN architectures underscores its strong transferability and generalizability. Furthermore, our model demonstrates significant efficiency gains, as it operates in less than half the time of baseline models on large graphs such as ArXiv and Products, while also boasting markedly reduced memory consumption.

## 2 PRELIMINARY

We focus on structural adversarial attacks for node classification. Consider a graph denoted as $\mathcal{G} = (A, X)$, where $A \in \{0, 1\}^{N \times N}$ represents its adjacency matrix, encompassing $m$ edges, and the node attributes are captured by $X \in \mathbb{R}^{N \times d_0}$. The central objective of structural adversarial attacks lies in the strategic manipulation of this graph by adding or deleting a limited number of edges, thereby causing a decline in the performance of GNNs applied to this altered structure. We primarily emphasize evasion attacks at test time, although it is worth noting that our methodology can also be extended to poisoning attacks during training (Zügner & Günnemann, 2019). Generally, we formulate the structural attack as follows:

$$\max_{\hat{A} \text{ s.t. } \|\hat{A} - A\|_0 < \Delta} \mathcal{L}(f_\theta(\hat{A}, X)), \tag{1}$$

where $\mathcal{L}$ represents the attack loss function, $\Delta$ denotes the attack budget, and $f_\theta(\cdot)$ is the GNN model applied to the graph. In the evasion attack scenario, the model parameter $\theta$ remains fixed throughout the adversarial attack process.

**Attack loss.** Empirically, the commonly used surrogate loss functions are Masked Cross Entropy (MCE) loss and Hyperbolic Tangent of the Margin loss (Geisler et al., 2021; Li et al., 2022). MCE loss is defined as $\mathcal{L}_{\text{MCE}} = \frac{1}{|\mathbb{V}^+|} \sum_{i \in \mathbb{V}^+} -\log(p_{i,c^*})$. Here, $\mathbb{V}^+$ denotes the set of correctly classified nodes, $c^*$ represents the ground-truth label of the node, and consequently, $p_{i,c^*}$ represents the probability or confidence score predicted for node $i$ after the softmax activation. The Hyperbolic Tangent of the Margin loss is given by $\mathcal{L}_{\text{tanhmargin}} = -\tanh(\mathbf{z}_{c^*} - \max_{c \neq c^*} \mathbf{z}_c)$, where $\mathbf{z}_c$ represents the logit predicting an arbitrary node as class $c$ before the softmax activation.

**Projected Randomized Block Coordinate Descent (PRBCD).** To better illustrate, we can reformulate the graph structural attack as follows:

$$\max_{P \text{ s.t. } P \in \{0,1\}^{N \times N}, \sum P \leq \Delta} \mathcal{L}(f_\theta(A \oplus P, X)), \tag{2}$$

where $p_{ij} = 1$ represents an edge flip. To conduct Projected Gradient Descent (PGD), we relax the discrete edge perturbation matrix to a continuous counterpart $P \in [0,1]^{N \times N}$ one. The operation $\oplus$ is defined as:

$$(A \oplus P)_{ij} = A_{ij} \oplus P_{ij} = \begin{cases} A_{ij} + P_{ij} & A_{ij} = 0 \\ A_{ij} - P_{ij} & A_{ij} = 1 \end{cases} \tag{3}$$

Previously, the gradient-based graph adversarial attacks calculate the gradient for all entries in the adjacency matrix, such as and PDG topology attack (Xu et al., 2019). It requires $O(N^2)$ gradient computations, making it suitable only to smaller datasets and impractical for larger graphs. While efforts have been made to circumvent the exhaustive calculation of complete gradients, such as GradArgmax (Dai et al., 2018) and SGA (Li et al., 2021), they each come with their own drawbacks. GradArgmax only computes gradients with respect to the connected edges in the adjacency matrix, allowing only for edge deletion with no edge addition, and thus leave the performance suboptimal. SGA extracts a subgraph comprising k-hop neighbors of the target node and flips edges with the largest gradient magnitudes within this subgraph. However, SGA is only suitable to the local attack scenario, restricting its capability in broader contexts.

To overcome these problems, Geisler et al. (2021) proposed PRBCD that only calculates the gradients of some certain places in the edge perturbation matrix $P$, referred to as the "block" or the "search space". In each step, the attack loss with continuous edge perturbation, denoted as $\mathcal{L}(f_\theta(A \oplus P, X))$ in Eq 2, is first calculated. Gradient descent is exclusively applied within the search space, and projection is then applied to constrain the updated values within the range of $[0,1]$. At the end of each step, only parts of the search space exhibiting the largest gradients are retained, while the remaining areas are discarded. Subsequently, the discarded search space is reconstructed by resampling from the edge perturbation matrix for the next epoch. Finally, after a few steps of gradient descent within the search space, PRBCD chooses the entries with the top-$\Delta$ gradients and flips the corresponding edges, where $\Delta$ is the total attack budget. As we can see from this full process of PRBCD attack, the memory requirements for maintaining gradients are reduced to the size of the active search space.

**Greedy Randomized Block Coordinate Descent (GRBCD).** GRBCD (Geisler et al., 2021) greedily flips the entries with the largest gradients in the block in each step until the perturbation budget is met. Consider the total number of steps to be $E$, $\sum_{t=1}^{E} \Delta_t = \Delta$. The budget is distributed evenly for each epoch. Compared to PRBCD, GRBCD exhibits a somewhat higher level of scalability because it continuously and greedily flips edges throughout the process, thus eliminating the necessity for the block size to exceed the total budget. Nonetheless, it is crucial to note that there is no assurance of how closely it approximates the actual optimization problem, and for effective performance, the block size should still be reasonably large.

As stated in Section 1, achieving better attack performance requires a large block size. Therefore, the time and memory consumption is still nontrivial involving larger datasets. Besides, the inherent unpredictability of randomized search space selection potentially introduces instability and suboptimal performance. As a matter of fact, for greedy attacks, the precise computation of gradient values is unnecessary. Therefore, our EGALA model focuses on identifying entries with the highest gradients quickly and accurately. EGALA can potentially search through all entries in the adjacency matrix while remaining scalable to large graphs.

## 3 METHODOLOGY

Given that the adjacency matrix inherently comprises a discrete $\{0,1\}^{N \times N}$ structure, in the context of greedy updates, the attacker's objective is to strategically flip edges associated with the highest gradient values. Precise gradient value calculations become unnecessary, as long as the attacker has a good identification of these key entries. We transform the gradient computation to an elegant closed-form solution and utilize clustering-based Approximate Nearest Neighbor Search (ANNS) to efficiently do the identification in the range of all entries in the adjacency matrix. Without the burden

of massive gradient computation, our model is scalable to larger graphs and avoid the instability of random block sampling. In each step, our model identifies entries with the most significant gradients in sublinear time relative to the total number of entries. We choose the top-$\Delta_t$ entries with the largest gradient values and greedily flip the corresponding edges, where $\Delta_t$ is the budget for the $t^{\text{th}}$ step.

The main contribution of our model lies in its capacity to efficiently pinpoint entries with the largest gradient values, being scalable to large graphs without the need of exact gradient computation or random block sampling. To illustrate our methodology, we start by introducing a surrogate model, an easy and useful representative of frequently used GNN models. Subsequently, we show that the gradient computation for this surrogate model has an elegant closed-form solution. Notably, this solution paves the way for the efficient identification of high-gradient entries, facilitated through the innovative application of ANNS techniques.

## 3.1 SURROGATE MODEL AND LOSS FUNCTION

We consider a practical attack setting, where attackers lack access to the target model, but can use the available training data to train a surrogate model and transfer the attacks on the surrogate model to the target model. This setting has been widely adopted in the literature of adversarial attacks to computer vision (Papernot et al., 2017; Wu et al., 2018; Tramèr et al., 2017) and graph neural networks (Zügner et al., 2018; Wang & Gong, 2019; Jin et al., 2021). Given the attacker's limited knowledge of the target model, it is customary to select the surrogate model from the most general GNN models. The prevailing choice for surrogate models for graph adversarial attack is Vanilla GCN (Kipf & Welling, 2016). Another commonly used surrogate model is Simplified Graph Convolutional Network (SGC), a linear variant of GCN proposed by Wu et al. (2019). SGC differs from GCN in that it replaces the nonlinear ReLU activation function between each layer with the identity function. This simplicity in computation makes SGC an attractive surrogate model for graph adversarial attacks, as evidenced in studies such as Li et al. (2021) and Zügner et al. (2018). In our EGALA model, we adopt a two-layer SGC as our surrogate model for attacks. Its formulation can be expressed as:

$$Z = \hat{A}(\hat{A}XW_0)W_1, \tag{4}$$

where $W_0 \in \mathbb{R}^{d_0 \times d_1}$, $W_1 \in \mathbb{R}^{d_1 \times d_2}$, $d_2$ equals to the number of classes. This can also be represented in a summation form:

$$\mathbf{z}_i = W_1^T \sum_{j=1}^{N} a_{ij}(W_0^T \sum_{k=1}^{N} a_{jk}\mathbf{x}_k). \tag{5}$$

For the sake of streamlined gradient computation, we opt for the hyperbolic tangent of the margin, as introduced in Section 2, as our surrogate loss function:

$$\mathcal{L} = \sum_{i \in \mathcal{S}} l_i = \sum_{i \in \mathcal{S}} - \tanh(\mathbf{z}_{i,c^*} - \max_{c \neq c^*} \mathbf{z}_{i,c}), \tag{6}$$

where $\mathbf{z}_{i,c}$ represents the logit predicting the node $i$ as class $c$, and $\mathcal{S}$ denotes the set of target nodes. In line with the implementation of Geisler et al. (2021), we set the target nodes to be all of the test nodes in the global attack scenario. For local attacks, we only need to specify it as a specific node.

## 3.2 GRADIENT COMPUTATION WITH CHAIN RULE

In this section, we show that with the surrogate model, the gradient of the entire adjacency matrix can be reduced to computing inner product of two $N$-by-$d$ matrices. This nice structure will then lead to an efficient routine for finding entries with top-$k$ gradient values.

**Chain Rule.** By applying the chain rule, we can decompose the gradient of the loss with respect to the adjacency matrix into three components. The derivative of the loss function with respect to the entry located at position $(p, q)$ in the adjacency matrix is represented as:

$$\frac{\partial \mathcal{L}}{\partial e_{pq}} = \sum_{i \in \mathcal{S}} \frac{\partial l_i}{\partial e_{pq}} = \sum_{i \in \mathcal{S}} \frac{\partial l_i}{\partial \mathbf{z}_i} \frac{\partial \mathbf{z}_i}{\partial a_{pq}} \frac{\partial a_{pq}}{\partial e_{pq}}, \tag{7}$$

where $e_{pq} \in \{0, 1\}$ indicates whether there exists an edge between node $p$ and node $q$, while $a_{pq}$ signifies the value of the element at row $p$, column $q$ in the normalized adjacency matrix. Then we can calculate the three components individually.

**Gradient of the Hyperbolic Tangent Loss.** The first term of equation 7 is the derivative of loss to the final layer output. For this term we need to consider the derivative of the hyperbolic tangent, which is given by $\nabla \tanh(x) = 1 - \tanh^2(x)$. Then we can get:

$$(\frac{\partial l_i}{\partial \mathbf{z}_i})_c = \begin{cases} -(1 - \tanh^2(\mathbf{z}_{i,c^*} - \max_{c \neq c^*} \mathbf{z}_{i,c})) & c = c_i^* \\ 1 - \tanh^2(\mathbf{z}_{i,c^*} - \max_{c \neq c^*} \mathbf{z}_{i,c}) & c = c_i^{\max} \\ 0 & \text{otherwise} \end{cases} = \begin{cases} -(1 - l_i^2) & c = c_i^* \\ 1 - l_i^2 & c = c_i^{\max} \\ 0 & \text{otherwise} \end{cases} \quad (8)$$

where $c_i^*$ represents the ground-truth label of the node, and $c_i^{\max} = \text{argmax}_{c \neq c^*} \mathbf{z}_{i,c}$.

**Gradient with respect to the Normalized Adjacency Matrix.** For the second component of equation 7, we have:

$$\frac{\partial \mathbf{z}_i}{\partial a_{pq}} = \begin{cases} a_{ip} W_1^T W_0^T \mathbf{x}_q & i \neq p \\ a_{ip} W_1^T W_0^T \mathbf{x}_q + W_1^T W_0^T \sum_{k=1}^N a_{qk} \mathbf{x}_k & i = p \end{cases} \quad (9)$$

Detailed calculations are provided in Appendix A. It's evident that $a_{ip} W_1^T W_0^T \mathbf{x}_q$ consistently appears in $\partial \mathbf{z}_i / \partial a_{pq}$, while $W_1^T W_0^T \sum_{k=1}^N a_{qk} \mathbf{x}_k$ is present only when $i = p$.

By merging the first two components, we can derive the gradient of the loss with respect to the normalized adjacency matrix, and we rewrite it in the form that separates the variables related to $p$ and $q$, so that it is feasible to extend to the matrix form:

$$\frac{\partial \mathcal{L}}{\partial a_{pq}} = \sum_{i \in \mathcal{S}} \frac{\partial l_i}{\partial \mathbf{z}_i} \frac{\partial \mathbf{z}_i}{\partial a_{pq}} = (\sum_{i \in \mathcal{S}} a_{ip} \frac{\partial l_i}{\partial \mathbf{z}_i} W_1^T)(W_0^T \mathbf{x}_q) + (\frac{\partial l_p}{\partial \mathbf{z}_p} W_1^T)(W_0^T \sum_{k=1}^N a_{qk} \mathbf{x}_k). \quad (10)$$

As we know from Equation 8, $(\frac{\partial l_i}{\partial \mathbf{z}_i})_c$ is non-zero only when $c = c_i^*$ or $c = c_i^{\max}$, and the absolute value is equal to $1 - l_i^2$. Therefore, we can simplify equation 10 as:

$$\begin{aligned} \frac{\partial \mathcal{L}}{\partial a_{pq}} = & (\sum_{i \in \mathcal{S}} a_{ip}(1 - l_i^2)[(W_1^T)_{c_i^{\max}} - (W_1^T)_{c_i^*}])(W_0^T \mathbf{x}_q) \\ & + ((1 - l_p^2)[(W_1^T)_{c_p^{\max}} - (W_1^T)_{c_p^*}])(\sum_{k=1}^N a_{qk} W_0^T \mathbf{x}_k), \end{aligned} \quad (11)$$

where $(W_1^T)_{c_i^{\max}}$ and $(W_1^T)_{c_i^*}$ refer to the $c_i^{\max}$-th and $c_i^*$-th row of the matrix $W_1^T \in \mathbb{R}^{d_2 \times d_1}$, respectively.

**Gradient to the Raw Adjacency Matrix.** The normalized adjacency matrix is defined as:

$$\bar{A} = \tilde{D}^{-\frac{1}{2}} \tilde{A} \tilde{D}^{-\frac{1}{2}} \quad (12)$$

where $\tilde{A} = A + I$, and $\tilde{D} = \text{diag}(\hat{d})$ is the diagonal matrix containing node degrees with a self-loop.

We can rewrite this in the following form:

$$a_{ij} = \frac{e_{ij}}{\sqrt{\hat{d}_i \hat{d}_j}} = e_{ij}(1 + \sum_{k \in \mathcal{N}_i} e_{ik})^{-\frac{1}{2}}(1 + \sum_{k \in \mathcal{N}_j} e_{jk})^{-\frac{1}{2}} \quad (13)$$

where $a_{ij} = (\bar{A})_{ij}$, $e_{ij} = (A)_{ij}$, $\hat{d}_i$ denotes the degree of node $i$ with a self-loop, and $\mathcal{N}_i$ denotes the set of neighbors of node $i$. Then we can derive the gradient of normalized adjacency matrix to the raw adjacency matrix as

$$\frac{\partial a_{ij}}{\partial e_{ij}} = (1 - \frac{e_{ij}}{2\hat{d}_i})\frac{1}{\sqrt{\hat{d}_i \hat{d}_j}}. \quad (14)$$

### 3.3 Gradient as a Matrix Product

Based on the derivations above, we can obtain an approximate closed-form solution to the gradient computation. As demonstrated in Equation 11, we observe that there are four $d_1$-dimensional vectors associated with the entry located at position $(p, q)$ in the adjacency matrix, two corresponding to node $p$ and the other two corresponding to node $q$. We proceed by constructing two matrices: $H_1 \in \mathbb{R}^{N \times d_1}$, where the $i^{\text{th}}$ row is given by $(1 - l_i^2)[(W_1^T)_{c_i^{\max}} - (W_1^T)_{c_i^*}]$, and $H_2 \in \mathbb{R}^{N \times d_1}$ with the $i^{\text{th}}$ row being $W_0^T \mathbf{x}_i$. Subsequently, we employ one-time neighborhood aggregation using $H_1$ and $H_2$ as latent feature matrices, resulting in $J_1$ and $J_2 \in \mathbb{R}^{N \times d_1}$. Notably, the neighborhood for aggregating $H_1$ comprises only the target nodes, whereas for $H_2$, it encompasses all nodes. Therefore, $\sum_{i \in \mathcal{S}} a_{ip}(1 - l_i^2)[(W_1^T)_{c_i^{\max}} - (W_1^T)_{c_i^*}]$ is the $p^{\text{th}}$ row of $J_1$ and $\sum_{k=1}^N a_{qk} W_0^T \mathbf{x}_k$ is the $q^{\text{th}}$ row of $J_2$. Consequently, we can readily express the gradient of the loss with respect to the normalized adjacency matrix in matrix form as:

$$\frac{\partial \mathcal{L}}{\partial \bar{A}} = J_1 H_2^T + H_1 J_2^T = [J_1, H_1] \cdot [H_2, J_2]^T, \tag{15}$$

where $[\cdot, \cdot]$ denotes the concatenation of two matrices along the columns.

The final step involves multiplying this gradient by the derivative of adjacency matrix normalization. To facilitate matrix multiplication, we also want to separates the variables related to $i$ and $j$ in the derivative, while as shown in Equation 14, the element of $e_{ij}$ is inseparable. However, considering the sparsity of edges in most graphs, where $e_{ij} \in \{0, 1\}$ and often takes the value zero, we can simplify the derivative as $\partial a_{ij} / \partial e_{ij} = 1/\sqrt{\hat{d}_i \hat{d}_j}$. Empirically, this approximation maintains good performance with our method.

We construct the reciprocal of square root degree vector, denoted as $\hat{d}_{\text{rsqrt}}$, where the $i^{\text{th}}$ element is $1/\sqrt{\hat{d}_i}$. We then scale each row of matrices $H_1, H_2, J_1, J_2$ by $\hat{d}_{\text{rsqrt}}$ to obtain the new matrices $\bar{H}_1, \bar{H}_2, \bar{J}_1, \bar{J}_2$. This modification enables us to derive the final expression for the entire gradient as the inner product of two $(N, 2d_1)$ matrices:

$$\frac{\partial \mathcal{L}}{\partial A} \approx [\bar{J}_1, \bar{H}_1] \cdot [\bar{H}_2, \bar{J}_2]^T. \tag{16}$$

We will leverage this special structure of the gradient to conduct efficient attack.

### 3.4 Clustering and Approximate Nearest Neighbor Search

Let's denote the matrices as follows: $Q_1 = [\bar{J}_1, \bar{H}_1]$, $Q_2 = [\bar{H}_2, \bar{J}_2] \in \mathbb{R}^{N \times 2d_1}$. To conduct attacks, the goal is to identify which entries in $Q_1 \cdot Q_2^T$ has the highest values. Although finding the exact top entries require $\mathcal{O}(N^2)$ time, thanks to the special matrix product structure, this problem can be reduced to the all-pair Approximate Nearest Neighbor Search (ANNS) or Maximum Inner Product Search (MIPS) problems (Ballard et al., 2015; Jiang et al., 2020; Shrivastava & Li, 2014; Malkov & Yashunin, 2018), where given a set of $N$ vectors, efficient algorithms have been developed to find the approximate top $k$ pairs in sublinear time. Theoretically, given a query $v$ a database with $N$ vectors $\{u_1, \ldots, u_N\}$, ANNS/MIPS algorithms are able to find $\arg\max_i u_i^T v$ in $\mathcal{O}(\log N)$ time (Friedman et al., 1977; Shrivastava & Li, 2014), so the overall cost of finding smallest pairs require only $O(N \log N)$.

However, many existing implementations and algorithms require substantial indexing time for nearest neighbor search, resulting in a significant amount of overhead at each iteration. Therefore, we implement a light-weighted ANNS algorithm in our attack based on the clustering idea proposed in (Jiang et al., 2020). First, we utilize the K-means algorithm to cluster the row vectors in matrices $Q_1$ and $Q_2$ separately, obtaining cluster centroids for each. Subsequently, we compare the cluster centroids of $Q_1$ with those of $Q_2$, calculating the inner products between these cluster centroids. This yields the $K$ closest cluster pairs. Next, we perform ANNS within each of these cluster pairs to obtain the indices of the $m$ closest vector pairs. We utilized the open-source library Faiss (Johnson et al., 2019) for this step. This step results in a total of $mK$ pairs of indices, corresponding to vectors with potentially high similarity. Finally, we compute the gradients associated with these index pairs using matrices $Q_1$ and $Q_2$. We select the index pairs with the highest inner product results and use them to identify the corresponding entries in the adjacency matrix for subsequent updates.

The pseudo code of our EGALA model is shown in Algorithm 1.

---

**Algorithm 1:** Efficient Gradient Approximation for Large-scale Graph Adversarial Attack (EGALA)

---

1   **Input**: graph $\mathcal{G} = (A, X)$, label $c^*$, GNN parameters $W_0, W_1$, loss function $\mathcal{L}$

2   **Parameter**: step size $\Delta_t$ for $t \in \{1, 2, ..., E\}$, the number of clusters: $w$, the number of closest cluster pairs: $K$, the number of closest vector pairs: $m$, period of cluster update: $f$, the number of samples for clustering: $n_s$

3   Initialize $\hat{A} \leftarrow A$

4   Randomly initialize the cluster centroids for the row vectors in matrices $Q_1$ and $Q_2$

5   **for** $t \in \{1, 2, ..., E\}$ **do**

6      Calculate matrix $Q_1$ and $Q_2$ according to 3.2 and 3.3

7      **if** $t \% f == 0$ **then**

8          **for** $Q_1$ *and* $Q_2$ **do**

9              Randomly select $n_s$ row vectors of the matrix

10              Conduct K-means and determine the $w$ cluster centroids

11              Cluster the $N$ row vectors of the matrix

12          **end**

13          Identify the $K$ closest cluster pairs by comparing the cluster centroids of $Q_1$ with those of $Q_2$.

14          **for** $k \in \{1, 2, ..., K\}$ **do**

15              Create a database containing row vectors from $Q_2$ that are associated with cluster pair $k$

16          **end**

17      **end**

18      **for** $k \in \{1, 2, ..., K\}$ **do**

19          Conduct ANNS with Faiss for row vectors of $Q_1$ with the database that belongs to the cluster pair $k$, and retrieve $m$ pairs of indices with the highest similarity scores

20      **end**

21      Compute the inner products of the row vectors in $Q_1$ and $Q_2$ that correspond to the index pairs

22      Update $\hat{A}$ by flipping $\Delta_t$ edges with the highest inner product values

23   **end**

24   Return $\hat{A}$

---

## 4   EXPERIMENTS

### 4.1   EXPERIMENTAL SETTING

**Datasets.** Our experiments are conducted on five commonly used graph datasets: Cora ML (Bojchevski & Günnemann, 2017), Citeseer (Giles et al., 1998), PubMed (Sen et al., 2008), ArXiv (Hu et al., 2020), and Products (Hu et al., 2020). Detailed information regarding these datasets is presented in Table 1. For Cora ML, Citeseer, PubMed, and ArXiv, we conduct our experiments on a standard 11GB GeForce GTX 1080 Ti, where most of the existing models can hardly scale to PubMed dataset (Geisler et al., 2021). For Products dataset, we use 48GB NVIDIA RTX A6000.

**Baseline models.** We conducted a thorough comparison of our model against state-of-the-art large-scale graph attack models, specifically PRBCD and GRBCD (Geisler et al., 2021). It is important to note that many other existing graph attack models are constrained by scalability issues (Li et al., 2021; Zügner et al., 2018), restricted on small datasets and local attacks. Geisler et al. (2021) have comprehensively compared various basic models in the global attack setting and demonstrated their superiority over models, including DICE (Waniek et al., 2018), PGD (Xu et al., 2019), and FGSM (Dai et al., 2018). As a result, we omit these baseline models from our comparative analysis and concentrate our evaluation on the two leading models, PRBCD and GRBCD.

**Attack settings.** In line with Geisler et al. (2021), we set the experiments as global attack, evasion attack, gray-box attack, and untargeted attack. For fair comparison, we take SGC (Wu et al., 2019) as the surrogate model for all the attacks in our experiments. For consistency with Geisler et al. (2021), we further compare our model with the baseline models using their originally suggested surrogate model, Vanilla GCN (Kipf & Welling, 2016).

Table 1: Statics of datasets.

| Dataset | # Nodes $N$ | # Edges | # Features $d_0$ | # Classes |
|---------|-------------|---------|------------------|-----------|
| Cora ML | 2,801 | 7,981 | 2,879 | 7 |
| Citeseer | 2,110 | 3,668 | 3,703 | 6 |
| PubMed | 19,717 | 44,324 | 500 | 3 |
| ArXiv | 169,343 | 1,157,799 | 128 | 40 |
| Products | 2,449,029 | 61,859,706 | 100 | 47 |

**GNN models.** We assess the effectiveness of our attack across a range of GNN models, including SGC, Vanilla GCN, Vanilla GDC (Gasteiger et al., 2019), Soft Median GDC (Geisler et al., 2021), Vanilla PPRGo (Bojchevski et al., 2020), and Soft Median PPRGo (Geisler et al., 2021).

**Hyperparameters.** In GNN training and attacks of PRBCD and GRBCD, we follow the settings in Geisler et al. (2021). We choose the block size $b$ of 1,000,000, 2,500,000, and 10,000,000 for Cora ML/Citeseer, PubMed and ArXiv/Products, respectively. For EGALA, we search the number of clusters $w$ in the set $\{5, 10, 20, 40\}$, the number of closest cluster pairs $K$ in $\{5, 10, 20, 40, 80\}$. We set the number of closest vector pairs $m$ for each cluster pair to be two or three times of the step size of each epoch. The period of cluster update $f$ is set as 2 or 3. The number of samples $n_s$ for clustering to be 500 and 1,000 for Cora ML/Citeseer and PubMed/ArXiv/Products, respectively.

### 4.2 EXPERIMENTAL RESULTS

We run each experiment with three random seeds. We set the attack budget to be $10\%$ of the total number of edges in the original graph. For smaller datasets like Cora ML and Citeseer, we also conduct a comparison between our EGALA, which utilizes clustering-based ANNS, and an alternative approach denoted as EGALA-N. EGALA-N directly computes $Q_1 \cdot Q_2^T$ to approximate gradients for all $N^2$ entries. This comparative analysis aims to demonstrate that our EGALA achieves results comparable to those of EGALA-N using direct gradient computation, thus reinforcing the effectiveness of our clustering-based ANNS approach.

**Attack performance.** The experiment results are shown in Table 2. In this table, the last column shows the original accuracy of the GNN models before the adversarial attack. "S-GDC" and "S-PPRGo" denote Soft Median GDC and Soft Median PPRGo, respectively. Upon scrutinizing the results, it becomes evident that our model consistently outperforms PRBCD and GRBCD across various datasets. Furthermore, our model exhibits strong transferability across various GNN architectures. The consistently superior performance of our approach across different GNN models highlights its versatility and ability to generalize. The comparative results with PRBCD and GRBCD using Vanilla GCN as the surrogate model is shown in Appendix B.2. To demonstrate the efficacy of our transfer attack approach, we also compare EGALA with adaptive PRBCD and GRBCD, the analysis shown in Appendix B.1. We conduct a thorough ablation study to assess the impact of different hyperparameters on EGALA and the results can be found in Appendix C.

**Time and memory cost.** The time and memory cost is presented in Table 3. As the dataset size increases, PRBCD and GRBCD necessitate larger block sizes for attacks, resulting in slower speeds and a substantial memory increase. In contrast, our method exhibits superior efficiency in terms of both time and memory usage, particularly excelling with large datasets, demonstrating the strong scalability of our method. It is noteworthy that the computational cost of EGALA on small datasets exceeds that of baselines, as well as our non-ANNS variant, EGALA-N. This increment in cost can be attributed to the ANNS methodology employed in EGALA, where Faiss is directly utilized. Exploring more efficient ANNS implementations holds the potential to notably diminish the overall computation cost. Importantly, for smaller datasets like Cora and Citeseer, the direct computation of matrix multiplication proves to be both rapid and incurs minimal memory cost. Conversely, for larger datasets like Arxiv and Products, the incorporation of the ANNS mechanism significantly mitigates computational costs, justifies its application.

Table 2: Node classification accuracy on various target GNNs using SGC as the surrogate model.

| Dataset | Model | PRBCD | GRBCD | EGALA | EGALA-N | Original |
|---------|-------|-------|-------|-------|---------|----------|
| Cora ML | SGC | 0.635±0.005 | 0.607±0.010 | 0.601±0.006 | **0.599±0.006** | 0.820±0.003 |
| | GCN | 0.644±0.010 | 0.623±0.005 | **0.615±0.006** | **0.615±0.006** | 0.821±0.008 |
| | GDC | 0.660±0.016 | **0.654±0.014** | 0.657±0.013 | 0.655±0.011 | 0.829±0.012 |
| | PPRGo | 0.699±0.009 | 0.716±0.008 | 0.697±0.011 | **0.695±0.010** | 0.820±0.011 |
| | S-GDC | 0.731±0.014 | 0.731±0.014 | 0.726±0.014 | **0.725±0.014** | 0.831±0.012 |
| | S-PPRGo | 0.769±0.011 | 0.772±0.013 | 0.768±0.016 | **0.766±0.014** | 0.811±0.013 |
| Citeseer | SGC | 0.561±0.004 | 0.517±0.003 | **0.516±0.005** | **0.516±0.006** | 0.713±0.005 |
| | GCN | 0.590±0.021 | 0.562±0.019 | 0.553±0.028 | **0.552±0.027** | 0.717±0.004 |
| | GDC | 0.589±0.002 | 0.573±0.014 | 0.570±0.004 | **0.571±0.008** | 0.705±0.012 |
| | PPRGo | 0.640±0.011 | 0.633±0.017 | **0.613±0.014** | **0.613±0.013** | 0.720±0.008 |
| | S-GDC | 0.643±0.008 | 0.633±0.015 | **0.627±0.011** | 0.628±0.011 | 0.715±0.009 |
| | S-PPRGo | 0.676±0.012 | 0.677±0.011 | **0.675±0.012** | 0.676±0.012 | 0.706±0.010 |
| PubMed | SGC | 0.608±0.009 | 0.569±0.012 | **0.564±0.010** | - | 0.772±0.006 |
| | GCN | 0.605±0.006 | 0.582±0.011 | **0.581±0.008** | - | 0.773±0.007 |
| | GDC | 0.637±0.014 | 0.611±0.019 | **0.607±0.017** | - | 0.776±0.006 |
| | PPRGo | 0.671±0.007 | 0.663±0.011 | **0.653±0.006** | - | 0.769±0.006 |
| | S-GDC | 0.671±0.014 | 0.649±0.021 | **0.641±0.014** | - | 0.776±0.007 |
| | S-PPRGo | 0.712±0.003 | 0.716±0.002 | **0.711±0.002** | - | 0.754±0.004 |
| ArXiv | SGC | 0.404±0.002 | 0.364±0.002 | **0.336±0.003** | - | 0.692±0.002 |
| | GCN | 0.426±0.006 | 0.421±0.001 | **0.402±0.010** | - | 0.698±0.004 |
| | GDC | 0.431±0.002 | 0.433±0.002 | **0.426±0.001** | - | 0.678±0.001 |
| | PPRGo | **0.502±0.004** | 0.515±0.008 | 0.505±0.008 | - | 0.675±0.007 |
| | S-PPRGo | 0.556±0.002 | 0.557±0.003 | **0.554±0.002** | - | 0.601±0.002 |
| Products | SGC | 0.453±0.001 | 0.313±0.002 | **0.302±0.002** | - | 0.656±0.001 |
| | GCN | 0.584±0.001 | 0.592±0.001 | **0.542±0.002** | - | 0.755±0.001 |
| | GDC | 0.569±0.002 | 0.563±0.001 | **0.561±0.001** | - | 0.665±0.001 |

Table 3: Time and memory cost of our attack versus the baselines.

| Dataset | Time per step (s) | | | | Memory (MB) | | | |
|---------|-------|-------|-------|---------|-------|-------|-------|---------|
| | PRBCD | GRBCD | EGALA | EGALA-N | PRBCD | GRBCD | EGALA | EGALA-N |
| Cora ML | 0.081 | 0.052 | 0.148 | **0.015** | 1101 | 1101 | 2,275 | **873** |
| Citeseer | 0.086 | 0.057 | 0.146 | **0.017** | 965 | 967 | 2,273 | **765** |
| PubMed | 0.207 | **0.125** | 0.173 | - | **1,735** | 1,817 | 2,606 | - |
| ArXiv | 1.064 | 1.020 | **0.229** | - | 6,723 | 6,315 | **4,616** | - |
| Products | 70.19 | 30.98 | **19.09** | - | 40,457 | 46,591 | **30,588** | - |

## 5 CONCLUSION

In this paper, we propose Enhanced Gradient Approximation for Large-scale Graph Adversarial Attack (EGALA), a novel approach to address the challenges associated with conducting graph adversarial attacks on large-scale datasets. By approximating gradients by a matrix product, coupled with the strategic application of clustering and Approximate Nearest Neighbor Search (ANNS), EGALA efficiently identifies entries with the largest gradients in the adjacency matrix without the need for explicit gradient calculations, and thus significantly enhances the model's scalability. Further, the idea of using ANNS for scalable graph attack can be used as a core building block for other structural attack algorithms in the future.

**Limitations.** The current paper focuses on the transfer attack setting since the derivation of gradient being a matrix product is based on the SGC network. It will be an interesting future direction to generalize this finding to general graph neural network architectures (probably with certain level of approximation) which will enable EGALA in the white-box attack setting. Further, beyond the clustering-based approach for ANNS used in the current implementation, it will be worthwhile investigating how other ANNS algorithms such as graph-based algorithms (Malkov & Yashunin, 2018; Chen et al., 2023; Jayaram Subramanya et al., 2019) and quantization-based algorithms Guo et al. (2020) can be applied to graph structural attacks.

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

## A  DETAILS OF GRADIENT COMPUTATION

Details of calculating $\frac{\partial \mathbf{z}_i}{\partial a_{pq}}$:

Case 1: $p \neq i$

$$\frac{\partial \mathbf{z}_i}{\partial a_{pq}} = W_1^T \frac{\partial}{\partial a_{pq}}[a_{ip}(W_0^T \sum_{k=1}^N a_{pk}\mathbf{x}_k)]$$
$$= W_1^T a_{ip} W_0^T \mathbf{x}_q = a_{ip} W_1^T W_0^T \mathbf{x}_q$$

Case 2: $p = i$:

$$\frac{\partial \mathbf{z}_i}{\partial a_{pq}} = \frac{\partial \mathbf{z}_i}{\partial a_{iq}}$$
$$= W_1^T \frac{\partial}{\partial a_{iq}}[a_{iq}(W_0^T \sum_{k=1}^N a_{qk}\mathbf{x}_j) + \sum_{j\neq q} a_{ij}(W_0^T \sum_{k=1}^N a_{jk}\mathbf{x}_k)]$$

a. $q \neq i$:

$$\frac{\partial \mathbf{z}_i}{\partial a_{pq}} = W_1^T \frac{\partial}{\partial a_{iq}}[a_{iq}(W_0^T \sum_{k=1}^N a_{qk}\mathbf{x}_k) + a_{ii}W_0^T \sum_{k=1}^N a_{ik}\mathbf{x}_k]$$
$$= W_1^T(W_0^T \sum_{k=1}^N a_{qk}\mathbf{x}_k + a_{ii}W_0^T \mathbf{x}_q)$$

b. $q = i$:

$$\frac{\partial \mathbf{z}_i}{\partial a_{pq}} = \frac{\partial \mathbf{z}_i}{\partial a_{ii}} = W_1^T \frac{\partial}{\partial a_{ii}}[a_{ii}(W_0^T \sum_{k=1}^N a_{ik}\mathbf{x}_k)]$$
$$= W_1^T(W_0^T \sum_{k=1}^N a_{ik}\mathbf{x}_k + a_{ii}W_0^T \mathbf{x}_i)$$

Therefore, for $p = i$ we have

$$\frac{\partial \mathbf{z}_i}{\partial a_{pq}} = W_1^T(W_0^T \sum_{k=1}^N a_{qk}\mathbf{x}_k + a_{ii}W_0^T \mathbf{x}_q)$$
$$= a_{ip}W_1^T W_0^T \mathbf{x}_q + W_1^T W_0^T \sum_{k=1}^N a_{qk}\mathbf{x}_k$$

## B  ATTACK PERFORMANCE COMPARISON WITH PRBCD AND GRBCD IN DIFFERENT SETTINGS

### B.1  COMPARATIVE EVALUATION OF EGALA WITH ADAPTIVE PRBCD AND GRBCD

To thoroughly understand the robustness of neural networks, it is essential to evaluate them against adaptive attacks, as these provide a more accurate reflection of their defensive capabilities (Mujkanovic et al., 2022; Tramer et al., 2020). Our proposed EGALA initially focuses on the transfer attack paradigm, leveraging a surrogate model based on a two-layer Simplified Graph Convolution (SGC). In this context, to demonstrate the efficacy of our transfer attack approach, we have drawn comparisons between our transfer EGALA and adaptive PRBCD and GRBCD, where in adaptive attacks the attacker has access to gradients from the victim model itself. Table 4 presents the results of these comparisons.

Table 4: Node classification accuracy comparing with adaptive PRBCD and GRBCD.

| Dataset | Model | PRBCD | GRBCD | EGALA | EGALA-N |
|---------|-------|-------|-------|-------|---------|
| Cora ML | GCN | 0.637±0.007 | **0.614±0.011** | 0.615±0.006 | 0.615±0.006 |
| | GDC | **0.638±0.014** | 0.697±0.005 | 0.657±0.013 | 0.655±0.011 |
| | S-GDC | **0.684±0.013** | 0.725±0.015 | 0.726±0.014 | 0.725±0.014 |
| Citeseer | GCN | 0.568±0.004 | **0.540±0.014** | 0.553±0.028 | 0.552±0.027 |
| | GDC | **0.552±0.008** | 0.599±0.020 | 0.570±0.004 | 0.571±0.008 |
| | S-GDC | **0.598±0.016** | 0.635±0.019 | 0.627±0.011 | 0.628±0.011 |
| PubMed | GCN | 0.615±0.004 | **0.569±0.007** | 0.581±0.008 | - |
| | GDC | **0.601±0.018** | 0.670±0.012 | 0.607±0.017 | - |

Table 5: Time and memory cost of adaptive attacks.

| Dataset | Model | Time per step (s) | | Memory (MB) | |
|---------|-------|-------|-------|-------|-------|
| | | PRBCD | GRBCD | PRBCD | GRBCD |
| Cora ML | GCN | 0.085 | 0.051 | 1071 | 1073 |
| | GDC | 0.158 | 0.093 | 1217 | 1239 |
| | S-GDC | 0.152 | 0.108 | 1275 | 1309 |
| Citeseer | GCN | 0.083 | 0.049 | 965 | 967 |
| | GDC | 0.124 | 0.074 | 1093 | 1073 |
| | S-GDC | 0.147 | 0.086 | 1193 | 1211 |
| PubMed | GCN | 0.200 | 0.128 | 1735 | 1817 |
| | GDC | 9.73 | 6.62 | 9985 | 10569 |

The results indicate that for adaptive attacks on Vanilla GCN, PRBCD and GRBCD display superior performance over transfer attacks, with GRBCD outperforming transfer EGALA and adaptive PRBCD, yielding the worst classification accuracy. For Vanilla GDC and Soft Median GDC models, adaptive PRBCD emerges as the most effective, surpassing attack results. Interestingly, adaptive GRBCD on these models does not outperform its transfer counterpart, which suggests that for greedy methods, being adaptive does not guarantee better performance than a transfer attack.

Additionally, we examined the computational resources required for these adaptive attacks. The results are shown in Table 5. We observed that the execution time and memory consumption dramatically increase when utilizing the more complex GNN models, constraining the scalability of adaptive attacks. Geisler et al. (2021) underscore this limitation by noting the absence of an efficient backpropagation-supported Personalized PageRank (PPR) implementation, which results in exorbitant computation and memory demands. Consequently, this restricts the feasibility of executing an adaptive, global attack on the Soft Median GDC to smaller datasets only.

In summary, while adaptive attacks can provide a more stringent test for model robustness, our EGALA provides competitive results in the transfer setting. Moreover, the limitations in resource requirements of adaptive attacks should not be overlooked, as they can be a restriction to testing model resilience in more complex, scalable real-world scenarios.

## B.2 COMPARATIVE ANALYSIS WITH VANILLA GCN AS THE SURROGATE MODEL FOR PRBCD AND GRBCD

We have further investigated the choice of surrogate models in graph adversarial attacks. We note that SGC's simplicity and scalability make it a prevalent choice for large graphs and is hence employed as a surrogate model in many state-of-the-art attack strategies, including ours (Li et al., 2021; Zügner et al., 2018). In order to have a more comprehensive experiment, we carried out additional experiments to consider PRBCD and GRBCD utilizing their originally suggested surrogate model, Vanilla GCN. We present the results of this analysis in Table 6.

Table 6: Node classification accuracy on various target GNNs with Vanilla GCN as the surrogate model for PRBCD and GRBCD.

| Dataset | Model | PRBCD | GRBCD | EGALA | EGALA-N |
|---------|-------|-------|-------|-------|---------|
| Cora ML | SGC | $0.641\pm0.008$ | $0.629\pm0.013$ | $0.601\pm0.006$ | $\mathbf{0.599\pm0.006}$ |
| | GCN | $0.637\pm0.007$ | $\mathbf{0.614\pm0.011}$ | $0.615\pm0.006$ | $0.615\pm0.006$ |
| | GDC | $0.659\pm0.010$ | $0.658\pm0.014$ | $0.657\pm0.013$ | $\mathbf{0.655\pm0.011}$ |
| | PPRGo | $\mathbf{0.695\pm0.011}$ | $0.719\pm0.007$ | $0.697\pm0.011$ | $\mathbf{0.695\pm0.010}$ |
| | S-GDC | $0.728\pm0.010$ | $0.735\pm0.011$ | $0.726\pm0.014$ | $\mathbf{0.725\pm0.014}$ |
| | S-PPRGo | $0.770\pm0.012$ | $0.777\pm0.014$ | $0.768\pm0.016$ | $\mathbf{0.766\pm0.014}$ |
| Citeseer | SGC | $0.583\pm0.015$ | $0.562\pm0.022$ | $\mathbf{0.516\pm0.005}$ | $0.516\pm0.006$ |
| | GCN | $0.568\pm0.004$ | $\mathbf{0.540\pm0.014}$ | $0.553\pm0.028$ | $0.552\pm0.027$ |
| | GDC | $0.583\pm0.011$ | $\mathbf{0.568\pm0.011}$ | $0.570\pm0.004$ | $0.571\pm0.008$ |
| | PPRGo | $0.642\pm0.004$ | $0.634\pm0.002$ | $\mathbf{0.613\pm0.014}$ | $\mathbf{0.613\pm0.013}$ |
| | S-GDC | $0.631\pm0.020$ | $0.631\pm0.023$ | $\mathbf{0.627\pm0.011}$ | $0.628\pm0.011$ |
| | S-PPRGo | $\mathbf{0.675\pm0.009}$ | $\mathbf{0.675\pm0.009}$ | $\mathbf{0.675\pm0.012}$ | $0.676\pm0.012$ |
| PubMed | SGC | $0.618\pm0.008$ | $0.584\pm0.012$ | $\mathbf{0.564\pm0.010}$ | - |
| | GCN | $0.615\pm0.004$ | $\mathbf{0.569\pm0.007}$ | $0.581\pm0.008$ | - |
| | GDC | $0.638\pm0.016$ | $0.611\pm0.022$ | $\mathbf{0.607\pm0.017}$ | - |
| | PPRGo | $0.671\pm0.011$ | $0.663\pm0.016$ | $\mathbf{0.653\pm0.006}$ | - |
| | S-GDC | $0.673\pm0.013$ | $0.651\pm0.020$ | $\mathbf{0.641\pm0.014}$ | - |
| | S-PPRGo | $\mathbf{0.711\pm0.001}$ | $0.717\pm0.003$ | $\mathbf{0.711\pm0.002}$ | - |

When comparing these results with those in Table 2, it is observed that substituting Vanilla GCN as the surrogate model for PRBCD and GRBCD does not markedly change their attack performance. Importantly, our EGALA maintains competitive performance relative to these baselines. The choice between Vanilla GCN and SGC as a surrogate appears to exert trivial influence on the overall performance of PRBCD and GRBCD, affirming the effectiveness of our EGALA across various experimental conditions.

## C ABLATION STUDY

To investigate the impact of various clustering parameters on the efficacy of our EGALA model, we conducted an ablation study using the Cora ML dataset. The attack model used is SGC (Wu et al., 2019). Our exploration centers on three key parameters: the number of clusters ($w$), the number of closest cluster pairs ($K$), and the period of cluster updates ($f$). We assess the implications on attack performance as well as resource utilization, including time and memory consumption, to gain a more comprehensive understanding. In this ablation study, we set the parameter governing the number of closest vector pairs ($m$) within each cluster pair to double the step size. Moreover, the number of samples ($n_s$) selected for clustering is consistently maintained at 500.

Figure 1 illustrates the effect of varying the number of clusters ($w$) on model performance, keeping the number of closest cluster pairs ($K$) at 20 and the update frequency ($f$) at 3. Figure 2 demonstrates how alterations to the number of closest cluster pairs ($K$) affect the model, with the number of clusters ($w$) held constant at 10 and the update frequency ($f$) at 3. Lastly, Figure 3 depicts the impact of different cluster update periods ($f$), given a fixed number of clusters ($w$) at 5 and closest cluster pairs ($K$) at 5.

The observed results indicate that a smaller number of clusters ($w$) leads to reduced computational time without compromising model performance. To ensure optimal performance, the cluster update frequency ($f$) must be relatively low, with a value of 3 appearing optimal, striking a balance between computational efficiency and attack efficacy. Variations in the model parameters shown in our tests have a negligible effect on memory usage. To summarize, the influence of clustering parameters on the EGALA model is minimal. Our analysis demonstrates that the model consistently delivers robust performance over a wide range of parameter configurations, highlighting its flexibility and stability under various settings.

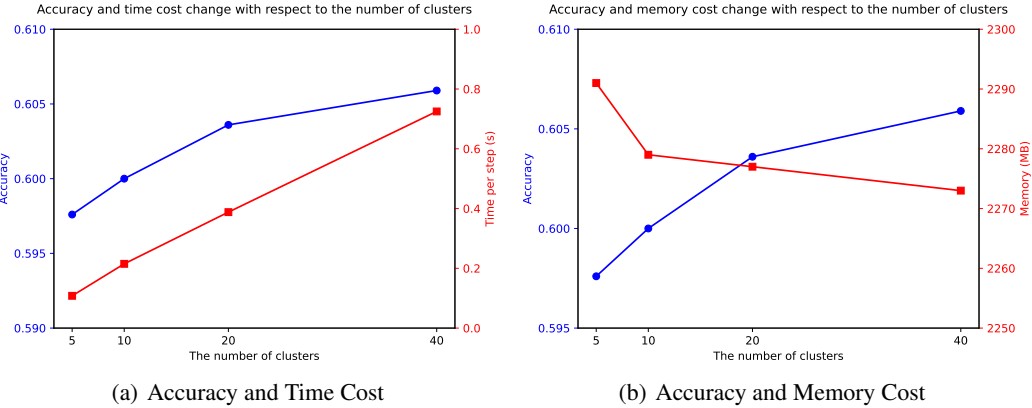

(a) Accuracy and Time Cost        (b) Accuracy and Memory Cost

Figure 1: Influence of the number of clusters $w$

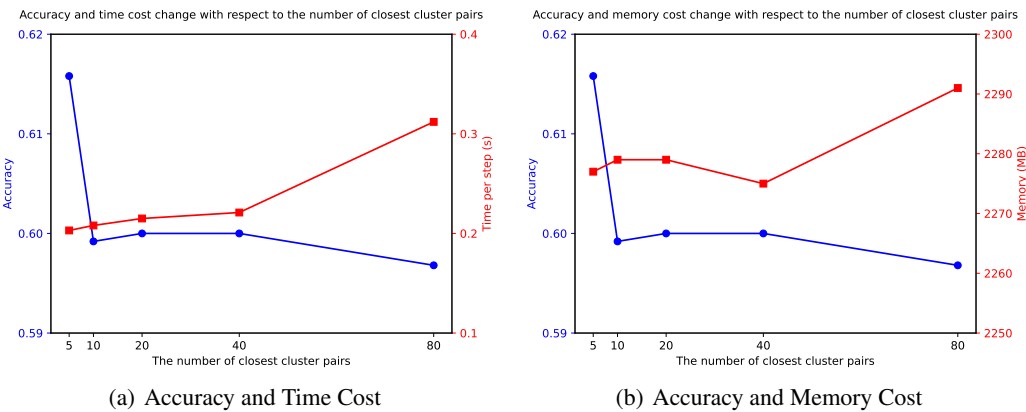

(a) Accuracy and Time Cost        (b) Accuracy and Memory Cost

Figure 2: Influence of closest cluster pairs $K$

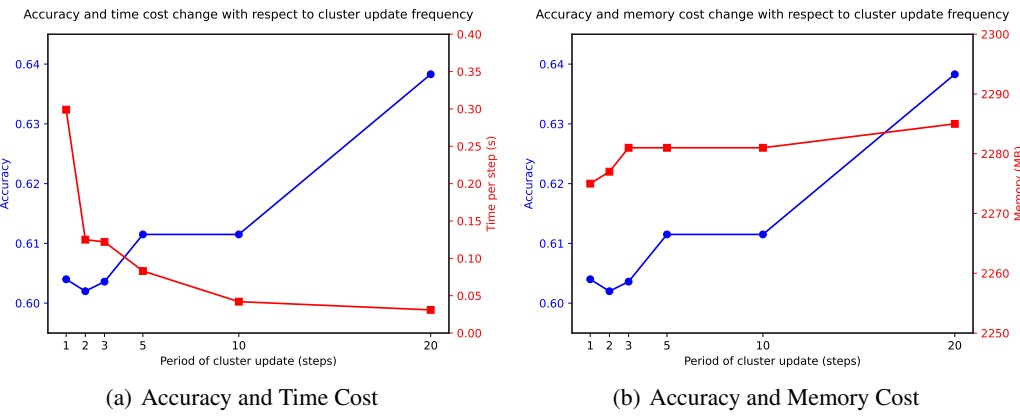

(a) Accuracy and Time Cost        (b) Accuracy and Memory Cost

Figure 3: Influence of the period of cluster update $f$

