# OpenReview forum: "EGALA: Efficient Gradient Approximation for Large-scale Graph Adversarial Attack"
_ICLR.cc/2024/Conference — Submitted to ICLR 2024_

### Official Review · Reviewer_BFci · 2023-10-29

**Soundness:** 2 fair
**Presentation:** 3 good
**Contribution:** 2 fair
**Rating:** 5
**Confidence:** 4

**Summary:**

This work proposes EGALA, a scalable graph adversarial attack based on gradient approximation. Specifically, authors exclusively focus on the SGC as the surrogate model for attacking. By formulating the gradient of loss with respect to adjacency matrix as matrix product, EGALA adopts a scalable nearest neighbor search algorithm to identify the edges with largest gradients. Experimental results indicate that EGALA is more effective and efficient than prior scalable attack methods.

**Strengths:**

- Overall, the paper is well-written.
- Addressing the scalability issue of graph adversarial attacks is important.
- Authors have evaluated on large-scale datasets.

**Weaknesses:**

- The gradient derivation is exclusively based upon the SGC model, which raises uncertainty about whether EGALA remains applicable to other, more advanced GNN models (e.g., GAT, GPRGNN, etc.). While authors have mentioned this limitation, I believe this is a critical issue and has to be addressed. Otherwise, I regret to say that the contribution of this work may not appear significant.
- There are some approximation steps in EGALA, such as Equation 16 and the nearest neighbor search. Given that authors have not provided the theoretical analysis on those approximation errors, it is less convincing whether EGALA indeed accurately identifies those edges with largest gradients. One way to address this concern could be comparing the gradients approximated by EGALA with the actual gradients on some small datasets.
- Authors only attack the scalable defense approach Soft Median. The results would be more compelling if authors could also attack other types of scalable defense methods (e.g., graph purification).

**Questions:**

- Have authors adopted mini-batch training on large graphs? How does the mini-batch training affect the gradient approximation in EGALA?
- How do authors perform hyperparameter tuning on all GNN models in the experiments?

---

> ### Author Response · Authors · 2023-11-21
>
> We appreciate the concern you have raised regarding the gradient derivation of our EGALA model and its applicability to more advanced GNN models such as GAT and GPRGNN. We fully recognize the importance of this issue. We have made some further studies.
>
> **[W1] Regarding the gradient derivation:**
> Although our proposed method only use SGC as the surrogate model, SGC is very useful for its simplicity, scalability and effectiveness.
> SGC is commonly used as surrogate model in graph adversarial attacks [1][2]. Also, the simplicity of SGC makes it even more applicable to large graphs. We have expanded our empirical evaluation, incorporating additional experiments to benchmark our EGALA model against adaptive PRBCD and GRBCD attacks. These new comparisons are detailed in the revised Appendix B.1. Using more advanced GNNs as surrogate models for attack does not necessarily improve the attack performance. Additionally, we examined the computational resources required conducting attacks on more advanced GNNs. Both time and memory costs rise significantly with the complexity of GNN models, constraining the scalability of adaptive attacks.
>
> In the future, we will try to find ways to apply our model to white-box setting, comparing the approximated gradients with actual ones, and evaluating on other defense methods.
>
> **[Q1] Mini-batch training on large graphs:**
> We adopt full-batch training on large graphs, which is the same as the baselines. Specifically, they chunk some operations (e.g. matrix multiplication) within the message passing step to successfully scale to larger graphs.
>
> **[Q2] Hyperparameter tuning:**
> To further explore parameter sensitivity and complexity, we have conducted additional ablation studies, presented in Appendix C, to analyze the influence of clustering parameters on our model's performance, both the accuracy and computational cost. For GNN models, we follow the training settings of the original paper of PRBCD and GRBCD.
>
> We appreciate the opportunity to address these critical aspects of our research and believe that the revisions have significantly strengthened the contribution of our work.
>
> [1] Li, Jintang, et al. "Adversarial attack on large scale graph." IEEE Transactions on Knowledge and Data Engineering, 2021.
>
> [2] Zügner, Daniel, et al. "Adversarial attacks on neural networks for graph data." Proceedings of the 24th ACM SIGKDD, 2018.

---

> > ### Comment · Reviewer_BFci · 2023-11-22
> > **Follow-up**
> >
> > Thanks for the response. However, most of my major concerns are still valid (e.g., gradient derivation and limited experiments on scalable defense methods). Thus, I keep my score unchanged.

---

### Official Review · Reviewer_d4tk · 2023-10-31

**Soundness:** 2 fair
**Presentation:** 3 good
**Contribution:** 2 fair
**Rating:** 5
**Confidence:** 3

**Summary:**

This paper proposes a graph adversarial attack method, EGALA, which is efficient and can be applied to large-scale graphs. The core idea of EGALA is to reconfigure the computation of loss gradients across the entire adjacency matrix as the inner product of two N-by-d matrices. Then, EGALA utilizes clustering and Approximate Nearest Neighbor Search (ANNS) to efficiently identify the entries with the most significant gradients in the adjacency matrix without the need for exact gradient computation, thus significantly enhancing the model’s scalability. The authors conduct comprehensive experiments across various datasets, demonstrating the effectiveness and transferability of EGALA.

**Strengths:**

1. The proposed idea is technically sound and seems novel to me.

2. The proposed method imposes minimal computational burden in terms of gradient calculations, making it highly efficient and memory-saving. It can be extended to larger graphs and avoids the instability associated with random block sampling.

3. The proposed method is easy to implement.

**Weaknesses:**

1. The proposed method uses SGC as the surrogate model and cannot be extended to other surrogate models. Additionally, in the experiments, the surrogate model used in the baseline is SGC, which may reduce the baseline's attack capabilities.

2. The experiments in the paper are not comprehensive enough. Providing more ablation experiments would be beneficial—for example, the impact of Δ_t in the algorithm. I also want to know the performance comparison of the PDG topology attack [1] and EGALA on small datasets.

3. The proposed method is applicable to attacks that only involve structural perturbations, limiting the method's applicability.

[1] Kaidi Xu, Hongge Chen, Sijia Liu, Pin-Yu Chen, Tsui-Wei Weng, Mingyi Hong, and Xue Lin. Topology attack and defense for graph neural networks: An optimization perspective. arXiv preprint

**Questions:**

Please see [Weaknesses] above.

---

> ### Author Response · Authors · 2023-11-21
>
> Thank you for your detailed and valuable feedback. Your input has led to substantive revisions and enhancements in our paper, and we are grateful for the opportunity to improve our work.
>
> **[W1] Regarding the extension to other surrogate models and the surrogate model of baselines:**
> Firstly, although our proposed method cannot be extended to other surrogate models, we demonstrate the importance of using SGC as our surrogate model for its simplicity, scalability and effectiveness.
> SGC is commonly used as surrogate model in graph adversarial attacks [1][2]. Also, the simplicity of SGC makes it even more applicable to large graphs.
> We have expanded our empirical evaluation, incorporating additional experiments to benchmark our EGALA model against adaptive PRBCD and GRBCD attacks. These new comparisons are detailed in the revised Appendix B.1. Using more advanced GNNs as surrogate models for attack does not necessarily improve the attack performance.
> Additionally, we examined the computational resources required conducting attacks on more advanced GNNs. Both time and memory costs rise significantly with the complexity of GNN models, constraining the scalability of adaptive attacks.
>
> Secondly, we carried out additional experiments to consider PRBCD and GRBCD utilizing their originally suggested surrogate model, Vanilla GCN. We present the results of this analysis in Appendix B.2.
>
> **[W2] Regarding the distribution of budgets:**
> In our experiemnt, the $\Delta_t$ we use is the same as that of GRBCD, where the budget is evenly distributed. This was done to ensure an equitable comparison of results. Furthermore, we acknowledge the potential for future work to explore more sophisticated budget distributions, as also suggested in the GRBCD original paper. Our current choice of $\Delta_t$ serves as a starting point, and we are open to investigating more complex allocation strategies as part of our continued research efforts.
>
> **[W2] Regarding the comparison with PDG:**
> We would like to clarify the rationale behind our baseline selection process. The PDG model has already been comprehensively compared with both PRBCD and GRBCD in their respective original publications. Consequently, we focused our comparative evaluation on the two more recent and advanced models: PRBCD and GRBCD. Nonetheless, we appreciate the importance of thorough baseline validation and are prepared to include PDG in our future evaluations to corroborate and extend the findings of the original papers.
>
> [1] Li, Jintang, et al. "Adversarial attack on large scale graph." IEEE Transactions on Knowledge and Data Engineering, 2021.
>
> [2] Zügner, Daniel, et al. "Adversarial attacks on neural networks for graph data." Proceedings of the 24th ACM SIGKDD, 2018.

---

> > ### Comment · Reviewer_d4tk · 2023-11-23
> > **Official comments**
> >
> > Thanks for your detailed response. The proposed method heavily depends on one model, weakening the contribution of this work. Thus, I am inclined to retain my score as it is.

---

### Official Review · Reviewer_rxwz · 2023-10-31

**Soundness:** 3 good
**Presentation:** 3 good
**Contribution:** 2 fair
**Rating:** 5
**Confidence:** 4

**Summary:**

The authors present EGALA, a method for constructing adversarial attacks on two-layer linear graph adversarial attacks at scale. EGALA approximates the gradient computation of the adjacency matrix as matrix product and efficiently identifies large entries in the gradients using Approximate Nearest Neighbor Search (ANNS), offering more scalable attacks with reduced memory and time consumption.

-- After rebuttal --

I agree that the more efficient transfer attack on a specialized 2-layer linear GNN still has some merit.  Therefore, I increase my score from 3 to 5. However, the comparison between transfer and adaptive attacks is quite counterintuitive. Deeper analyses and more comprehensive comparisons will be needed for a future version.

**Strengths:**

1. The derivation of the approximating gradient is convincing and elegant. The paper provides a solid theoretical analysis of how to derive the gradient of loss with respect to the adjacency matrix as a simple matrix product.

2. The proposed EGALA improves the efficiency of naive attacks without sacrificing the attack capability in the evaluated transfer attack setting. The author approximates the gradient with a matrix product and leverages the acceleration techniques, ANNS algorithm, to further improve the efficiency. The motivation and design mechanism of this method is sound.

**Weaknesses:**

1. EGALA is limited to attacking the surrogate model of two-layer linear GCN (essentially 2-layer SGC), and it can be only applied in the transfer attack setting when the victim model is not the same as the surrogate model. However, it has been shown in [1] that the transfer attack is much weaker than the adaptive attack, and the robustness evaluated under transfer attacks exhibits a strong false sense of security. This concern significantly weakens the contribution of this work.

2. The major baseline PRBCD is a randomized block coordinate method. PRBCD is efficient by selecting a small block size. More importantly, it is a general attack algorithm that can be applied to potentially any GNN model, without being limited to two-layer linear GCNs. Overall, the advantages of EGALA over PRBCD and GRBCD are not convincing enough. First, the attack performances of EGALA, PRBCD, and GRBCD are comparable in the transfer attack, while it is expected that PRBCD and GRBCD will provide much stronger adaptive attacks, especially when the evaluated model is robust GNNs (although no such study is presented). Second, the time complexity depends on many hyperparameters such as block size and number of clusters. However, there is no discussion and ablation study on the hyperparameter setting of baselines such as PRBCD and GRBCD. Therefore, the reported time cost comparison is not convincing enough.

3. There is a lack of time complexity comparison of EGALA and EGALA-N. It will be better to provide detailed analysis as well as corresponding ablation experimental results. In Table 3, the paper shows that EGALA and EGALA-N share the same time and memory cost,  which raises concerns about the advantage of the proposed clustering-based ANNS. Additionally, it is unclear why the cost of EGALA on PubMed is higher than the other baselines.

4. Lack of comprehensive ablation studies on several components, e.g., clustering method, number of clusters and number of closest vector pairs, period of cluster update. These components or hyperparameters can influence the accuracy and computation cost. Ablation studies should be included to show the impact of each technical component.


[1] Mujkanovic, Felix, et al. "Are Defenses for Graph Neural Networks Robust?." Advances in Neural Information Processing Systems 35 (2022): 8954-8968.

**Questions:**

Please refer to the weakness.

---

> ### Author Response · Authors · 2023-11-21
>
> Thank you for your thorough and insightful comments. Your constructive feedback has played a crucial role in the enhancement of our manuscript. We have undertaken a series of thoughtful modifications and additions in our work to address your points. Moreover, there are certain clarifications we would like to present regarding aspects of our research.
>
> **[W1 \& W2] Regarding the adaptive attack:**
> We expanded our empirical evaluation, incorporating additional experiments to benchmark our EGALA model against adaptive PRBCD and GRBCD attacks. These new comparisons are detailed in the revised Appendix B.1. Our results indicate that while adaptive attacks on the Vanilla GCN indeed demonstrate effectiveness, adaptive GRBCD does not consistently outperform its transfer attack, particularly within more complex GNN models like Vanilla GDC. The adaptive GRBCD fails to surpass its transfer counterpart in these contexts, suggesting that the advantage of adaptivity in greedy methods is non-universal and may not be decisive in all scenarios. Additionally, we examined the computational resources required for these adaptive attacks. We found that both time and memory costs rise significantly with the complexity of GNN models, constraining the scalability of adaptive attacks. This finding is important as it underscores the trade-offs between the intensity of attack methodologies and their practical deployability, especially with more elaborate GNN architectures.
>
> **[W2 \& W3] Regarding the computational cost:**
> We would like to apologize for any confusion caused by our initial presentation of the computational cost comparison. We have revised Table 3 in Section 4.2 to seperate the computational analysis of both EGALA and EGALA-N. The the increment of computational cost of EGALA on small datasets can be attributed to the ANNS methodology employed in EGALA. Exploring more efficient ANNS implementations holds the potential to notably diminish the overall computation cost. More importantly, for smaller datasets like Cora and Citeseer, the direct computation of matrix multiplication proves to be both rapid and incurs minimal memory cost. Conversely, for larger datasets like Arxiv and Products, the incorporation of the ANNS mechanism significantly mitigates computational costs, justifies its application.
>
> When comparing the computational cost of PRBCD and GRBCD, we used the recommended hyperparameter settings in the original papers to ensure fidelity in our comparative analysis. We also studied the influence of block size on the performance of the baseline. The results are shown in the following table.
>
> | Block size | $10^3$ | $10^4$ | $10^5$ | $10^6$ | $10^7$ |
> | --- | ---: | ---: | ---: | ---: | ---: |
> | Accuracy | 0.645 | 0.630 | 0.622 | 0.616 | 0.619 |
> | Time per step (s) | 0.014 | 0.014 | 0.015 | 0.051 | 0.195 |
> |Memory (MB) | 649 | 735 | 797 | 1101 | 2097 |
>
>
> **[W4] Ablation studies:**
> We conducted ablation studies, presented in Appendix C, to analyze the influence of clustering parameters on our model's performance, comparing both the accuracy and computational cost simultaneously.
>
> Your feedback has been instrumental in propelling our research forward. We appreciate the opportunity to address these critical aspects of our research and believe that the revisions have significantly strengthened the contribution of our work.

---

> > ### Comment · Reviewer_rxwz · 2023-11-22
> > **Further concerns**
> >
> > While I have not yet fully evaluated all of the points in your response, I would like to quickly express my unsolved concerns about the adaptive attack.
> >
> > I appreciate the additional experiments on the comparison of the transfer attack of your EGALA approach and the adaptive attack of PRBCD and GRBCD. However, these results are not fully convincing due to the following reasons.
> >
> > (1) The graph structure attack has been shown to transfer well when the surrogate model and victim model have similar architectures, such as similar graph convolutions. Therefore, it is not surprising that transfer attacks can still attack GCN and GDC in the provided experiments.
> >
> > However, the failure of transfer attacks and the false sense of security mainly happen when the victim model and surrogate model have intrinsic differences [1]. For instance, the surrogate model is vanilla GCN but the victim model is a robust GNN with new graph convolutions like SoftMedian [1] or ElasticGNN [2]. This is why I wonder how well the proposed transfer attack works for robust GNNs in my initial comment. In fact, there is no evaluation of how well the proposed algorithm attacks any robust GNNs.
> >
> > (2) It is unclear why the adaptive PRBCD and GRBCD can not consistently outperform the transfer attacks. For the chosen datasets like Cora and CiteSeer, it is feasible to select a large block size to cover the full-gradient case, which avoids any randomness or approximation error for gradient estimation. More studies and analyses on their differences need to be justified.
> >
> > [1] Are Defenses for Graph Neural Networks Robust? NeurIPS 2022
> >
> > [2] Elastic Graph Neural Networks, ICML 2021

---

> > > ### Author Response · Authors · 2023-11-22
> > >
> > > Thank you for your feedback! I appreciate the opportunity for further discussion.
> > >
> > > (1) I would like to clarify our evaluation process related to robust GNNs. In our paper, we indeed conducted experiments that evaluate the transfer attacks on robust GNNs. As you correctly pointed out, the distinction in architecture between vanilla and robust GNNs can significantly influence the success rate of transferability. To address this concern, we evaluate transfer attacks on robust GNN models. Specifically, we examined our proposed EGALA against models like SoftMedian GDC and SoftMedian PPRGo, which are designed to withstand adversarial attacks. The results of these evaluations are presented in Table 2 of our paper. The results indicate the efficacy of our proposed method, providing insights into its impact on models with inherent robustness features.
> > >
> > > (2) As indicated in our experimental results, we observe that adaptive GRBCD do not consistently outperform transfer attacks on the datasets examined, such as Cora and CiteSeer, where the corresponding results are detailed in Tables 2 and 4 of our paper. Our findings are consistent with those reported in the original paper[1], where the corresponding results are detailed in Tables F.4 and F.5. To address your point concerning the selection of block size, we carefully selected a block size of $10^6$ for our experiments on both Cora and CiteSeer, following the recommendations from the original paper[1]. We ensure the use of a block size that does not compromise the model performance, and our experimental design avoids the introduction of additional variables that could skew the comparison between methods. We believe that the empirical evidence presented is intended to serve as a reference point for the performance of EGALA relative to existing methods. Nevertheless, we recognize the value of such discussions and would be open to exploring these comparisons in greater depth in future work.
> > >
> > > Thank you once again for your constructive comments, and we hope that the aforementioned clarifications address any concerns regarding our experimental design and findings.

---

> > > > ### Comment · Reviewer_rxwz · 2023-11-22
> > > > **Thanks**
> > > >
> > > > Thanks for reminding me that some robust GNNs are included. Could you please provide some explanation or intuition as to why adaptive attacks can not outperform transfer attacks? It is quite unusual according to existing research and my experience. BTW, I will be happy to reevaluate my score after the reviewer discussions.

---

> > > > > ### Author Response · Authors · 2023-11-23
> > > > >
> > > > > Thank you for pointing out. Generally it is expected that adaptive attacks would outperform transfer attacks, but there are indeed scenarios where this may not be the case. Currently it is really hard for us to say why exactly that adaptive GRBCD does not outperform its transfer counterpart in some cases. This phenomenon provides an interesting direction for future exploration and analysis.

---

### Official Review · Reviewer_ca4y · 2023-10-31

**Soundness:** 3 good
**Presentation:** 3 good
**Contribution:** 3 good
**Rating:** 5
**Confidence:** 4

**Summary:**

The authors propose a novel strategy, termed EGALA, for performing an adversarial attack on graph neural networks w.r.t. the discrete graph structure. For this, the authors utilize an efficient approximate method for determining the elements with the largest gradient in the N x N adjacency matrix (where N is the number of nodes). The authors compare their method to the state-of-the-art attacks PRBCD and GRBCD.

**Strengths:**

1. In contrast to the state-of-the-art PRBCD and GRBCD, the novel approach EGALA does not rely on randomly sampled candidate edges to achieve efficiency. Instead, EGALA relies on approximate nearest neighbor search (with randomization) to focus always on the important edges.
1. EGALA is 3.5 times faster than PRBCD and 1.5 faster than GRBCD on the large products graph. The memory cost is about 30% smaller.
1. In the presented experiments, EGALA quite consistently outperforms PRBCD and GRBCD in terms of attack strength, although, the differences are often small.

**Weaknesses:**

1. The empirical evaluation is not exhaustive. E.g., the authors should evaluate also local attacks or visualize the approximation of the gradient (for small graphs). This would make the work more convincing in regard of general applicability as well as that the approximation is sensible.
1. The authors only consider a grey box setting where the perturbations are transferred between models. This setting certainly has its merits. However, as pointed out previously, it is vital to assess neural networks with adaptive attacks [I, II] to get a proper estimate of the model's robustness. The authors should have prominently placed disclaimers and a comprehensive discussion on for what purpose the attack could be used.
1. The attack is model specific and thus, it is not straightforward to make it "adaptive" for other GNNs than SGC.
1. In connection to 2 & 3, the authors should craft experiments where they compare their transfer EGALA with an adaptive PRBCD and GRBCD. For example, the authors could attack defenses like Jaccard GCN, or SVG GCN (see [I]).
1. The authors do neither test nor discuss local attacks on larger graphs like Papers100M (like PRBCD/GRPCB did).

Minor:
1. The authors could improve the references from Sec. 3.3. to eq. 11

[I] - On Adaptive Attacks to Adversarial Example Defenses, Carlini et al., NeurIPS 2020
[II] - Are Defenses for Graph Neural Networks Robust?, Mujkanovic et al., NeurIPS 2022

**Questions:**

1. What is the exact asymptotic complexity of the approach?
1. How is the computational cost affected by the hyperparameters?
1. Is it necessary to approximate the derivate d a_ij / d e_ij for scalability?

I will raise the score if the questions and other points are addressed accordingly.

---

> ### Author Response · Authors · 2023-11-21
>
> Thank you for your thorough and constructive feedback. Your insights have been invaluable in refining our paper. Based on your suggestions, we have implemented several modifications and enhancements to our work.
>
> **[W2, W3, W4] Regarding the adaptive attack:**
> We have expanded our empirical evaluation, incorporating additional experiments to benchmark our EGALA model against adaptive PRBCD and GRBCD attacks. These new comparisons are detailed in the revised Appendix B.1. Our results indicate that while adaptive attacks on the Vanilla GCN indeed demonstrate effectiveness, adaptive GRBCD does not consistently outperform its transfer attack, particularly within more complex GNN models like Vanilla GDC. The adaptive GRBCD fails to surpass its transfer counterpart in these contexts, suggesting that the advantage of adaptivity in greedy methods is non-universal and may not be decisive in all scenarios. Additionally, we examined the computational resources required for these adaptive attacks. We found that both time and memory costs rise significantly with the complexity of GNN models, constraining the scalability of adaptive attacks. This finding is important as it underscores the trade-offs between the intensity of attack methodologies and their practical deployability, especially with more elaborate GNN architectures.
>
> **[Q2] Computational cost affected by hyperparameters:**
> To further explore parameter sensitivity and complexity, we have conducted additional ablation studies, presented in Appendix C, to analyze the influence of clustering parameters on our model's performance, both the accuracy and computational cost. Also, it is noteworthy that the primary contributor to computational load in EGALA on small datasets is the Approximate Nearest Neighbor Search (ANNS). As such, pursuing more advanced ANNS methods is likely to reduce computational costs substantially.
>
> **[Q3] The approximation of $\frac{\partial a_{ij}}{\partial e_{ij}}$:**
> The approximation of the derivative $\frac{\partial a_{ij}}{\partial e_{ij}}$ is necessary and indispensable. Empirically, the attack performance degrades if we ignore this part. Moreover, according to Equation (14) of our paper, $\frac{\partial a_{ij}}{\partial e_{ij}}$ is inversely proportional to the squared root of node degrees. By attacking edges connected to lower-degree nodes, the model incurs a higher misclassification rate, which is consistent with recent findings in [1]. The attack performance without the approximation of $\frac{\partial a_{ij}}{\partial e_{ij}}$ is shown in the following table.
>
> | Model | SGC | GCN | GDC | PPRGo | S-GDC | S-PPRGo
> |---|---:|---:|---:|---:|---:|---:|
> | EGALA | 0.599 | 0.615 | 0.655 | 0.695 | 0.725 | 0.766 |
> | No $\frac{\partial a_{ij}}{\partial e_{ij}}$ | 0.780 | 0.788 | 0.824 | 0.798 | 0.824 | 0.797 |
>
> In the future, we will conduct more experiments to evaluate on local attacks, visualizing the approximated gradients, finding ways to apply our model to white-box setting, and evaluating on other defense methods. Your comments have been instrumental in improving our study and the clarity of our findings. We appreciate the opportunity to address these critical aspects of our research and believe that the revisions have significantly strengthened the contribution of our work.
>
> [1] Li, Kuan, et al. "Revisiting graph adversarial attack and defense from a data distribution perspective." The Eleventh International Conference on Learning Representations, 2022.

---

> > ### Comment · Reviewer_ca4y · 2023-11-23
> > **Follow up**
> >
> > I thank the authors for addressing the posed weaknesses and questions of the presented work.
> >
> > It is interesting that GRBCD performs worse in the adaptive setup. Perhaps a more extensive hyperparameter search is required for it to perform on par or better. Nevertheless, the adaptive PRBCD seems to be substantially stronger in the adaptive case. Having said this, it appears to be a misleading presentation to put the adaptive results in the appendix and they should be incorporated into the main part (e.g., Table 2).
> >
> > Again, I absolutely agree that the grey box setting has its merits (e.g. computational complexity). Nevertheless, the authors should make clear that EGALA is not a substitute for adaptive/whitebox attacks. I stand to my opinion that
> > > The authors should have prominently placed disclaimers and a comprehensive discussion on for what purpose the attack could be used.
> >
> > Perhaps also elaborating on the trade-offs mentioned in the author's rebuttal.
> >
> > Regarding d a_ij / d e_ij.  It would be helpful to place this "assumption" more prominently in section 3. And include the discussion of the rebuttal.
> >
> > In summary, I think there are merits to this work and the availability of multiple adversarial attacks is vital for an adaptive evaluation of a method. Moreover, for this, it would be beneficial to highlight the differences in the found solutions / perturbed adjacency matrices in comparison to P/GRBCD.
> >
> > Since the requested discussion of grey box vs. adaptive attacks is not reflected in the updated paper and the results using adaptive attacks are deferred to the appendix, I am inclined to retain my score as it is.

---

### Meta-Review · Area_Chair_VBih · 2023-12-06

**Metareview:**

The authors propose a novel strategy, termed EGALA, for performing an adversarial attack on graph neural networks. However, the method can only use SGC as the proxy model for graph attacks. Considering the various types of GNNs for different graph datasets in graph learning, I agree other reviewers point out that the methods' contributions are somehow limited. Therefore, I decide to reject this paper.

**Justification For Why Not Higher Score:**

Only limited in linear models. The contributions are limited.

**Justification For Why Not Lower Score:**

N/A

---

### Decision · Program_Chairs · 2024-01-16

Reject